



# Impact of Flood Control Systems on the Probability Distribution of Floods

Salvatore Manfreda[1], Domenico Miglino[1], Cinzia Albertini[1,2]

[1]Dipartimento di Ingegneria Civile, Edile e Ambientale, Università degli Studi di Napoli Federico II, 80125 Napoli, Italy;
[2]Dipartimento di Scienze Agro Ambientali e Territoriali, Università degli Studi di Bari Aldo Moro, 70126 Bari, Italy;

*Correspondence to*: Salvatore Manfreda (salvatore.manfreda@unina.it)

**Abstract.** Detention dams are one of the most effective practices for flood mitigation. Therefore, the impact of these structures on the basin hydrological response is critical for flood management and the design of flood control structures. With the aim to provide a mathematical framework to interpret the effect of flow control systems on river basin dynamics, the functional
relationship between inflows and outflows is investigated and derived in a closed-form. This allowed the definition of a theoretically derived probability distribution of the peak outflows from in-line detention basins. The model has been derived assuming a rectangular hydrograph shape with a fixed duration, and a random flood peak. In the present study, the undisturbed flood distribution is assumed to be Gumbel distributed, but the proposed mathematical formulation can be extended to any other flood-peak probability distribution. A sensitivity analysis of parameters highlighted the influence of detention basin
capacity and rainfall event duration on flood mitigation on the probability distribution of the peak outflows. The mathematical framework has been tested using for comparison a Monte Carlo simulation where most of the simplified assumptions used to describe the dam behaviours are removed. This allowed to demonstrate that the proposed formulation is reliable for small river basins characterized by an impulsive response. The new approach for the quantification of flood peaks in river basins characterised by the presence of artificial detention basins can be used to improve existing flood mitigation practices, support
the design of flood control systems and flood risk analyses.

## 1 Introduction

During the last decades, the growing number of hydrological extremes have raised economic losses and risk perception at the global scale (Peduzzi, 2005; Di Baldassarre et al., 2010; Winsemius et al., 2015). The impact of natural disasters has been quantified in a recent study by UNISDR (2015), which is based on the Emergency Events Database (EM-DAT). According to
EM-DAT, flooding impacted on nearly 2.5 billion people in the period 1994-2013. A more recent study by Munich Re (NatCatSERVICE,2020) reported 3798 flash flood events that produced economic losses of about 592 billion USD and killed around 100.000 people worldwide during the last two decades (2000–2018).

In this context, climate change and anthropic activities are probably accelerating the number of extremes (Fisher & Knutti, 2016; Papalexiou and Montanari, 2019). In fact, these two factors are significantly modifying river basin hydrology (Di



Baldassarre et al., 2017), which now requires renewed methods and techniques. Therefore, the need to find appropriate measures for flood hazard prevention and mitigation is becoming more and more pressing.

Flood risk management is a complex challenge for hydrologists that need to identify dynamic solutions for flood mitigation. Plate (2002) underlined the need to update flood mitigation plans according to the changing conditions (e.g., climate, populations, land use). Hence, the design of flood mitigation systems requires rapid methodologies to evaluate scenarios and

intervention measures and obtain effective flood risk management strategies (Franzi et al., 2016).

One of the most effective action for the attenuation of peak discharges is represented by a detention basin system or coordinated reservoir operations (e.g., UDFCD, 2016; Jacob et al., 2019; Seibert et al., 2014). The proper design of a detention basin is an extremely complex task given the large number of factors that are involved in the decision process: technical engineering features of the structure, site selection and environmental characteristics. For this reason, Bellu et al. (2016) proposed an

innovative method to optimize the dimensioning and site selection of a flood mitigation system. The method follows three steps that include a preliminary sizing, a site location and optimization according to environmental objectives. Nevertheless, the sizing of the detention volume is based, in most of the cases, on a given design flood event neglecting the random nature of the phenomena and the impact of the structure on a large spectrum of events.

The flood attenuation caused by the presence of artificial reservoirs is influenced by hydrologic and hydraulic factors, such as

flood wave shape and duration, as well as by the storage capacity and geometric parameters of the detention basin. Therefore, it is crucial to build a mathematical scheme able to interpret the functional relationships, even in a simplified form, among the mentioned variables.

With this aim, the present manuscript introduces a Theoretically Derived probability Distribution (TDD) of detention basins outflows, which is obtained assuming the incoming flood peaks randomly distributed and characterized by rectangular

hydrographs of fixed duration (see i.e., Manfreda et al., 2018). In the present case, flood distribution is assumed to be Gumbel for the sake of simplicity, but the proposed approach can be applied to any flood-peak probability distribution. Performances of the proposed method are investigated under different configurations and are tested with a numerical simulation of flood mitigation. This method can be applied to improve existing flood mitigation approaches, support the design of flood control systems, flood risk and damage analyses.

This paper is organized as follow: in Section 2, the conceptual scheme of the detention basin adopted to derive the mathematical formulation of outflows is described. Furthermore, the hydraulic concepts and assumptions to analytically compute the derived probability distribution of the peak outflows are introduced; in Section 3, the proposed methodology is tested under different reservoir configurations and compared with the results of the numerical simulations and a brief description of the numerical model is also provided; and in Section 4 main findings and results are discussed.





## 2 The conceptual scheme

In-line detention dams (also known as flow-through dams) are constructed solely with the purpose of flood control and mitigation of flood risks in downstream communities and ecosystems. Unlike reservoir dams, which are primarily built for water storage or power generation, the spillway (opening) is located at the same height as the riverbed level, allowing the river to continue its natural flow under normal conditions. When water levels rise above the spillway, the dam restricts the amount flowing through the opening, decreasing peak flow. Since detention dams minimally affect rivers natural flows, under normal conditions negative environmental and socioeconomic impacts, such as sediment accumulation, restriction of water flow to downstream communities and ecosystems, and breaching during very extreme flood events, can be minimized or avoided altogether.

The schematization of the detention dam has been simplified with the aim to obtain a mathematical description of the outflows associated with a given hydrograph. In particular, we assumed that the dam body has two openings: a low-level opening at the basement and a crest spillway. The former is assumed to let pass the flow below a given control value, while the latter starts functioning only when the volume of the dam is completely filled up to the crest level. Thereafter, the water starts to flow from both the openings, and the control is mainly exerted by the basin volume above the emergency spillway (crest level). This scheme can be described in closed form that may help the construction of a derived probability distribution of the outflow from a detention dam.

### 2.1 The hydraulic characteristics of the problem

Let's first introduce the key equations controlling the dynamics of a system like the one under study. The first equation to introduce is represented by the stage-storage capacity curve which is able to describe the morphology of the gorge closed by a specific dam. The function is generally represented by a power-law:

$$W(h) = w_1 \, h^n, \tag{1}$$

where W(h) [$L^3$] is the water storage of detention basin, $w_1$ [$L^{3-n}$] is the parameter of the stage-storage capacity curve, h [L] is the water level in the reservoir and n [-] is the exponent influenced by the shape of the control volume. The exponent ranges between 1 and 4.5, where 1 is associated with a prismatic geometry with vertical surrounding walls and 4.5 is associated with a more complex morphology closed by more gentle lateral slopes.

The streamflow of the river system reaching the reservoir is altered by it based on the hydraulic characteristics of the dam. The outflow will be controlled by the amount of water accumulated in the system according to the continuity equation

$$\frac{dW(t)}{dt} = Q_{in}(t) - Q_{out}(t), \tag{2}$$

where $Q_{in}(t)$ is the incoming flux and $Q_{out}(t)$ is the outflow from the reservoir.

The outflow can be computed using the traditional formulation of hydraulics based on the variation of the water level, h, in the reservoir, which can be derived from the stage-storage capacity curve (Equation (1)) and continuity equation (Equation (2)). In particular, we can assume that the outflow for a simple scheme with two openings (the low-level opening and the emergency



spillway) and the characteristics described in Figure 1 may vary according to four different discharge laws expressed as a function of the reservoir water level stage. The outflow can be described as follow:

$$Q_{out} = \begin{cases} 0 < h \le q_f \Rightarrow 0 \\ \mu_s b\sqrt{2g}(h - q_f)^{\frac{3}{2}}, q_f < h \le h_f \\ \mu_f A\sqrt{2g}(h - h_f)^{\frac{1}{2}}, \ h_f < h \le h_s \\ \mu_f A\sqrt{2g}(h - h_f)^{\frac{1}{2}} + \mu_s L\sqrt{2g}(h - h_s)^{\frac{3}{2}}, \ h > h_s \end{cases}, \qquad [3]$$

where, $\mu_f$ [-] is the coefficient of discharge of the low-level opening at the basement (suggested values may range between 0.6 (assuming a thin sharp-edged) and 0.8 (assuming a wall with thickness two times larger than the opening height), $A$ [L²] is the area of the low-level opening (i.e., the product of the opening width, $b$ [L], and the minimum dimension between the top of the flow surface at the opening exit and the bottom of the opening), $\mu_s$ [-] is the coefficient of discharge of the spillway crest (suggested values range around 0.3-0.4 based on the geometry of the weir), $q_f$ [L] is the height of the low-level opening at the

basement, $L$ [L] is the effected crest length, $h_f$ [L] is the height of the barycentre of the low-level opening, $h_s$ [L] is the height of the spillway crest, and $g$ [LT⁻²] is the acceleration due to gravity.

The above expression includes the three main configurations that may occur in the proposed scheme with the increase of the water level stage in the reservoir. A detailed description of the scheme proposed along with graphical indication of the parameters meaning is given in Figure 1. In particular, the opening at the bottom does not exert a significant control on the

incoming flow as long as the water level does not generate a submergence of the opening (i.e., as long as $h \le h_s$). The water flow starts to be limited when the water stage reaches the value $h_f$, after which the opening is submerged. This allows to fill the storage volume of the reservoir up to the level of the crest spillway. After this stage, the reservoir tends to operate a mitigation that is influenced by the water storage capacity of the reservoir above the crest level and the hydraulic characteristics of the spillway. An example of flood mitigation obtained via numerical simulation is given in Figure 2 which provides a

comparison between a synthetic hydrograph and the outflow from a detention dam.

For the scope of the present study, the outflow has been simplified through the following set of equations:

$$Q_{out} = \begin{cases} Q_{in}, \ 0 < h \le h_f \\ Q_c, \ h_f < h \le h_s \\ \mu_f A\sqrt{2g}(h - h_f)^{\frac{1}{2}} + \mu_s L\sqrt{2g}(h - h_s)^{\frac{3}{2}}, \ h > h_s, \end{cases} \qquad [4]$$

where $Q_c$ is the control value of discharge that is computed using the discharge equation of the submerged opening and assuming $h = h_s$.

Assuming a rectangular hydrograph of the incoming flow, it is possible to derive the peak flow associated with an incoming flood peak. Following the simplifying assumption given in Equation (4), the outflow is not affected by the presence of the dam for lower streamflow values, while it is modified when the inflow exceeds the control discharge. In particular, the outflow remains almost constant as long as the reservoir is filled and, thereafter, it is controlled by the crest spillway. In this last





configuration, it is possible to use the linear reservoir concept for the water volume accumulated above the elevation of the

crest spillway.

In order to estimate the peak flow associated with a specific rectangular hydrograph of constant discharge equal to $Q_{max}$, we should recall the expression of the peak flow generated by a simple linear reservoir, which can be described as follow:

$$Q_{lam} = Q_{max}(1 - e^{-t_p/k}).$$   [5]

where $t_p$ [T] is the event duration, and $k$ [T] is the delay constant of the conceptual linear reservoir.

The above equation should be modified, in the present case, considering that the flood event should fill the dam water storage capacity (or detention basin) before reaching the crest spillway. Therefore, the crest spillway will be activated only after a time

$t_{filling} = W_{max}/(Q_{max} - Q_c)$,

where $W_{max}$ is the volume of water accumulated in the dam at the crest level $h_s$.

When the volume below the spillway crest is totally filled, the crest spillway starts functioning for discharge values above the

control discharge that is released at the bottom. Therefore, the peak outflow, $Q_{p,out}$, assumes the following form:

$$Q_{p,out} = Q_c + (Q_{max} - Q_c)(1 - Exp[-\left(t_p - \frac{W_{max}}{(Q_{max}-Q_c)}\right)/k_{eq}]) ,$$   [6]

where $k_{eq}$ [T] is the equivalent delay constant of the conceptual reservoir associated with the outflow. This parameter can be derived exploiting the characteristics of the spillway and the stage-storage capacity curve. In particular, according to the linearity concept, the two functions should have the same exponent. Under such a hypothesis, the parameter $k_{eq}$ can be

estimated as:

$$k_{eq} = \frac{w_2}{\mu_s L\sqrt{2g}} ,$$   [7]

where $w_2$ is the coefficient of the rescaled stage-storage capacity curve above the crest level. Such parameter should be computed in order to get the best approximation of the function describing the volumes above the mentioned level, $h_s$, imposing a coefficient $n$ equal to 1.5 for the rescaled stage-storage capacity curve (referred to the stage-storage capacity curve

above the crest level). With this aim, the parameter can be computed comparing the two functions and setting that they are equal in a point $h_m$ which is representative of the range of variability of the water level above the crest level.

$$w_2 = \frac{W_1(h_s+h_m)^n - W_1 h_s^n}{h_m^{1.5}}.$$   [8]

Within the present manuscript, we assumed the parameter $h_m$ was set equal to $h_s$.

These assumptions allowed to derive a functional relationship between the reservoir inflow and outflow, which can be used to

invert the function respect to the incoming flow and associate a probability to each flow discharged value by exploiting the theory of derived distributions (Benjamin and Cornell, 2014).



### 2.1.1. Estimation of the event duration

The assumption of a rectangular hydrograph may produce a significant overestimation of the flood volume. Therefore, the parameter $t_p$ should be defined accounting for the real volume associated with a realistic flood hydrograph. In this contest, we

can rely on the Flow Duration Frequency Reduction curve (FDF) proposed by the NERC (Natural Environment Research Council, 1975), which describes the maximum average discharge $q(D)$ as a function of the event duration $D$:

$$q(D) = Q_{max} e^{-\frac{D}{\omega}},$$  [9]

where $\omega$ represents the characterising basin time response that is frequently associated with the lag-time of the river basin. Adopting the above formulation, Fiorentino (1985) suggested a simple form of synthetic hydrograph redistributing the volume

symmetrically respect to the time of the peak. This leads to the following form of hydrograph

$$q(t) = Q_{max} e^{-2\frac{|t|}{\omega}}.$$  [10]

Based on the above formulation, it is possible to impose that the duration $t_p$ of the equivalent rectangular event has the same volume of the synthetic hydrograph of Equation (10) in the temporal window of $\omega$ around the peak flow. This led to the following equivalence:

$$t_p = \frac{(e-1)}{e}\omega \cong 0.632\,\omega\ .$$  [11]

Given the above assumption, the term $t_p$ will be named equivalent event duration from now on.

### 2.2. Functional relationship between the incoming discharge and the outflow

The possibility to identify the analytical relationship between two processes where one represents the stochastic forcing allows to determine the derived probability distribution of the variable. This approach has been used several times for flood maxima

(Eagleson, 1972; De Michele and Salvadori, 2002; Gioia et al., 2008); soil moisture (Rodriguez-Iturbe and Porporato, 2007; Manfreda and Fiorentino, 2008); and scour process (Manfreda et al., 2018). In the present case, the methodology has been applied to the laminated flood peak.

Given the above approximations, it is possible to derive the inverse function of the peak discharge function of the peak outflow. This equation can be obtained mathematically inverting Equation (6) and exploiting the parametrization introduced above. The

inverse function assumes the following form:

$$Q_{max} = g^{-1}(Q_{p,out}) =$$

$$= -\frac{k_{eq}^2 Q_c e^{\frac{t_p}{t_p+k_{eq}}} - k_{eq} W_{max} e^{\frac{t_p}{t_p+k_{eq}}} - t_p W_{max} e^{\frac{t_p}{t_p+k_{eq}}} - (k_{eq}^2 + k_{eq}\,t_p)\,Q_{p,out} e^{\frac{t_p}{k}}}{k_{eq}^2\left(e^{\frac{t_p}{k_{eq}}} - e^{\frac{t_p}{t_p+k_{eq}}} + \frac{t_p}{k_{eq}} e^{\frac{t_p}{k_{eq}}}\right)}.$$  [12]

This expression can be used to analytically compute the derived probability distribution of the peak outflow from a detention dam characterized by a storage capacity $W_{max}$, an equivalent delay constant $k_{eq}$, invested by flood hydrograph of equivalent





event duration $t_p$. With this aim, any probability distribution of the flood peaks can be used given the monotonic nature of the above expression. The expression of the TDD will be (see Benjamin and Cornell, 2014):

$$f_y(y) = \left|\frac{dg^{-1}(x)}{dy}\right| f_x(g^{-1}(x)),$$  [13]

where the derivative of $g^{-1}(Q_{p,out})$ assumes the following form:

$$\frac{dg^{-1}(Q_{p,out})}{dQ_{p,out}} = \frac{k_{eq}e^{\frac{t_p}{k_{eq}}} + t_p e^{\frac{t_p}{k_{eq}}}}{k_{eq}\left(e^{\frac{t_p}{k_{eq}}} - e^{\frac{t_p}{t_p + k_{eq}}} + \frac{t_p}{k_{eq}} e^{\frac{t_p}{k_{eq}}}\right)}.$$  [14]

In order to describe the probability distribution of the outflows, we should divide it according to the three potential configurations of the detention dams: 1) undisturbed flow; 2) accumulation of water in the reservoir; 3) activation of the crest spillway. Based on these assumptions, the probability distribution of the outflow is subdivided into three components and modelled by the following set of equations:

$$p(Q_{p,out}) = \begin{cases} p_{Q_{max}}(Q_{p,out}), & Q_{p,out} < Q_c \\ \int_{Q_c}^{\frac{W_{max}}{t_p} + Q_c} p_{Q_{max}}(Q_{p,out})\, dq, & Q_{p,out} = Q_c \\ \left|\frac{dg^{-1}(Q_{p,out})}{dQ_{p,out}}\right| f_{Q_{p,out}}(g^{-1}(Q_{p,out})), & Q_{p,out} > Q_c \end{cases}$$  [15]

Equation (15) describes the general form of the probability distribution of the outflow from a detention dam, where the first component coincides with the distribution of the incoming flow as long as it is below the control discharge of the lower opening ($Q_{p,out} < Q_c$). Assuming that the lower opening is able to control the outflow around $Q_c$ after submergence, there is a mass probability in $Q_c$ depending on the storage volume of the reservoir ($Q_{p,out} = Q_c$). After these two phases, the outflow is affected by the lamination due to the water volume accumulation above the crest level ($Q_{p,out} > Q_c$).

## 3. Applications

### 3.1. Examples of application of the TDD of the detention basin outflows under different configurations

In order to explore the behaviour of the proposed formulation, we investigated the effects of different parametrization on the derived distribution starting from a single distribution of floods. In Figure 3, the influence of the storage capacity and hydrograph duration on the outflow of the reservoir is shown. We depicted the probability density functions (pdfs) associated

with increasing storage capacity obtained by raising the crest level from two meters up to eight meters (moving top-down in the figure) and considering two distinct equivalent event durations of 30 minutes and one hour. As expected, hydrographs with larger duration tend to saturate sooner the water storage capacity of the reservoir, reducing also the lamination effects. On the other hand, the increase in water storage capacity leads to a proportional growth of flood peak mitigation. These graphs describe the behaviour of a reservoir providing an output consistent with the dynamics of the process.





In Figure 4, we modified the coefficient of the stage-storage capacity curve, $w_1$, using the values of 5000 and 10000 and explored equivalent event durations ranging from half an hour to two hours. Graphs display how the pdfs of outflows are altered by the presence of a dam with these characteristics. In the present example, the impact of the lower opening can be better appreciated with a mass probability around $Q_c$ that is equal to 52m³/s. It must be clarified that the cross-section of the opening has been increased in this second example on purpose to emphasize its impact on the proposed mathematical scheme.

**3.2. Testing the reliability of the proposed method**

In the current work, a numerical simulation of flood mitigation through a detention basin with the characteristics reported in Figure 1 was carried out using the same forcing adopted for the proposed TDD. Therefore, we adopted the Gumbel distribution as reference distribution to generate random values of discharge and numerically simulate the dynamics of the detention dam and its peak outflows. It must be clarified that any probability distribution of floods (e.g., generalised extreme value, three-

parameter log-normal, generalised logistic and Gumbel distributions) can be applied. An example of the numerical simulation is given in Figure 2 with the consequent attenuation of the hydrograph due to the detention dam.

The numerical simulation was carried out with the main scope of testing the theoretically derived probability distribution of laminated peak flows and also quantify the impact of the approximations adopted to obtain a closed-form of the solution. Therefore, the comparison of the theoretically derived distribution and numerical outflows helps understanding the reliability

of the proposed methodology. Results are given in the following graphs.

Figure 5 provides a comparison of different pdfs obtained modifying the maximum water storage capacity of the dam and its height. It can be noted that the values of the probability distribution replicate fairly well those obtained with the numerical simulations. The approximation of a fixed discharge from the submerged opening induces a small dispersion of values around the control value of discharge, $Q_c$, that the theoretical probability distribution is not able to capture. Moreover, the adopted

approximations slightly overestimate the outflows which can be interpreted as a safety approximation for flood mitigation planning.

Comparing the different pdfs, it should be clarified that the parameters have been changed looking for combinations of dam heights, $h_s$, and coefficient $w_1$ leading to similar water storage capacity on each row. This allows to demonstrate that it is much more effective to increase the area flooded by the reservoir (the parameter $w_1$ represent the rate of increase of the water storage

with the water level) rather than increase the height of the dam.

Last analysis performed is given in Figure 6 where we tested the performances of the TDD assuming the lower opening closed. Such a condition may be representative of an ordinary dam with an assigned flood retention volume or used for water supply purposes, where the volume above the crest spillway leads to a lamination of floods. This configuration has been compared with the scheme described in the previous sections that includes the presence of the lower opening. The difference between

the two configurations is given in Figure 6.A (closed opening) and Figure 6.B (lower opening active). The two graphs show





the ability of the mathematical formulation to properly interpret also the present configuration, offering a wide spectrum of potential applications in hydraulic design.

Finally, in order to test the impact due to the approximation adopted by the rectangular hydrographs, we also compared the results of a numerical simulation where the hydrograph is assumed to be a symmetric exponential one according to the

expression given in Equation (11). The comparison is given in Figure 7 for three different equivalent event durations. It can be appreciated how the use of an equivalent event duration allowed to reproduce correctly the flood mitigation that looks very similar to those obtained with a symmetric exponential hydrograph. It must also be underlined that with the increase of the duration of the event such an approximation tends to deteriorate the result of the proposed model.

## 4. Conclusion

The present manuscript introduces a new formulation useful to quantify the impact of detention dams on the probability distribution of floods. We must acknowledge that the formulation was obtained with several simplifying assumptions that include the shape of the incoming hydrograph; the approximation used to interpret the flow through the lower opening; the approximation of a linear reservoir for the flow above the crest level. Summing all these, it is really satisfying to see that the obtained formulation can fairly well interpret the dynamics of such hydraulic infrastructures providing an analytical description

of the impact of artificial reservoirs on flood dynamics. This may be extremely useful in properly addressing the effects of water infrastructures on floods. The TDD can be used for detention dams, but the formalism can also be applied to ordinary dams just setting the control discharge to zero and assigning a given value of the water level in the dam. Therefore, the formalism is versatile and can be applied in different contexts. The strongest assumption is represented by the rectangular hydrograph which can be realistic for small river basins characterised by relatively small concentration time. With the aim to

minimize the impact of such a choice, we adopted an equivalent event duration in the formulation that allowed to account for the flood variability during a specific event. However, this assumption may become limitative in large river basins where hydrograph evolves over large areas and its shape is also not simple to be predicted. In fact, large river basins may display complex hydrographs with multiple peaks that require a specific approach.

The proposed method may be used in some contexts such as the projects and design of small lamination dams and detention

dams realized in small river basins. For instance, there are several river basins along the coastline, which drain a high amount of water in short durations, affecting cities and towns developed along the waterfront. These areas are typically exposed to frequent flood events that may impose the need to properly identify potential solutions for flood mitigation.

The scheme can be used to carry out preliminary dimensioning of these structures and eventually could be coupled with other tools to identify optimal configurations for flood mitigations. Moreover, the scheme can be applied to any probability

distribution of floods including the case of floods that are already subject to laminations allowing the description of a scheme of nested dams.



This topic is still under investigation and its study will be applied to identify optimal solutions in flood control systems quantifying the impact of structure on the full spectra of floods.

**Notations**

| | | |
|---|---|---|
| 265 | $\alpha$ [-] | Scale parameter of Gumbel distribution |
| | $\beta$ [-] | Location parameter of Gumbel distribution |
| | $\mu_f$ [-] | Coefficient of discharge of the submerged low-level opening |
| | $\mu_s$ [-] | Coefficient of discharge of the crest spillway |
| | $A$ [m$^2$] | Area of the low-level opening |
| 270 | $b$ [m] | Width of the low-level opening rectangular section |
| | $d$ [m] | Height of the low-level opening |
| | $g$ [m$^2$/s] | Acceleration due to gravity |
| | $h$ [m] | Variable water level within the detention basin |
| | $h_m$ [m] | Mean of water levels over the spillway crest |
| 275 | $h_s$ [m] | Height of the spillway crest |
| | $h_f$ [m] | Height of the barycentre of the low-level opening |
| | $k$ [s] | Storage coefficient of the linear reservoir method |
| | $k_{eq}$ [s] | Equivalent delay constant of the conceptual reservoir associated with the outflow |
| | $L$ [m] | Effected crest length |
| 280 | $n$ [-] | Exponent of the stage-storage capacity curve |
| | $p(Q)$ [-] | Probability density function of outflows |
| | $q_f$ [m] | Height of the low-level opening |
| | $Q_c$ [m$^3$/s] | Design outflow from the low-level opening |
| | $Q_{in}$ [m$^3$/s] | Inflow in the detention basin |
| 285 | $Q_{max}$ [m$^3$/s] | Peak flow incoming in the detention basin |
| | $Q_{out}$ [m$^3$/s] | Outflow from the detention basin |
| | $Q_{p,out}$ [m$^3$/s] | Peak outflow from the detention basin |
| | $t$ [s] | Time |
| | $t_{filling}$ [s] | Time after which the crest spillway starts functioning |
| 290 | $t_p$ [s] | Equivalent flood duration; |
| | $\omega$ [s] | Lag-time of the river basin |
| | $W_{max}$ [m$^3$] | Water storage capacity at the crest level; |





| | |
|---|---|
| $W$ [m$^3$] | Variable storage capacity of the detention basin |
| $w_1$ [m$^{3-n}$] | Parameter of the stage-storage capacity curve |
| $w_2$ [m$^{3/2}$/s] | Parameter of the equivalent stage-storage capacity curve |

**Acknowledgement**

This paper is part of a project entitled "Hydraulic risk mitigation in coastal basins with in-line expansion tanks: an integrated sizing approach" was funded by the Italian Ministry of Environment, Land and Sea.

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





# List of figures

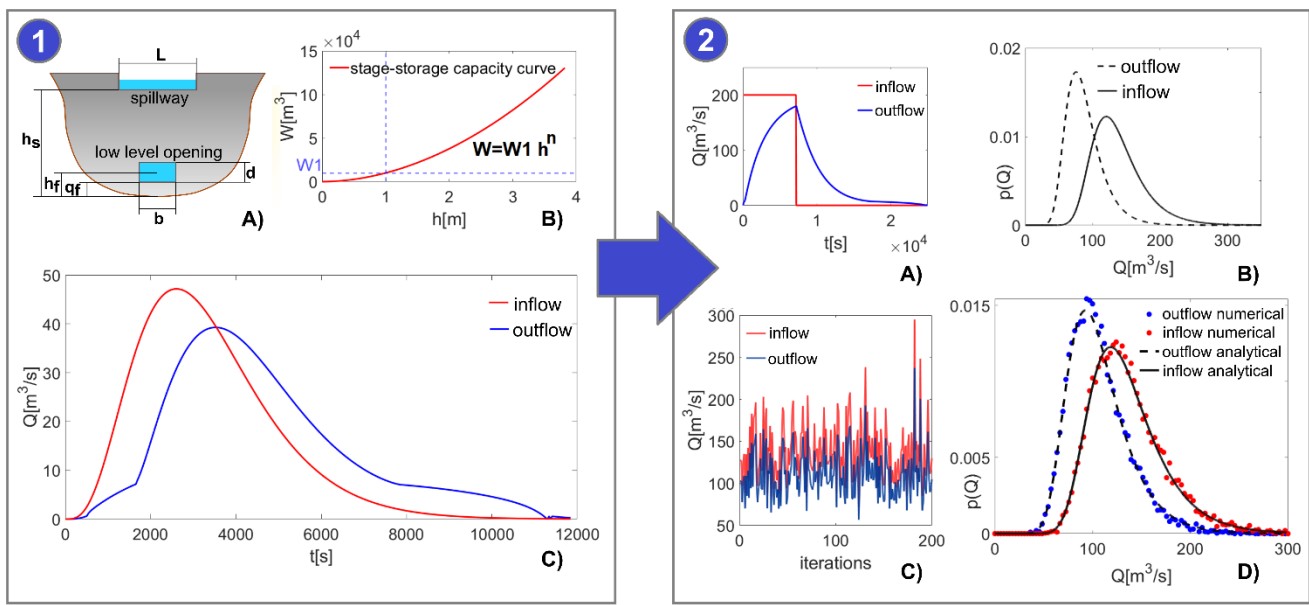

*Figure 0. Graphical abstract, illustrating the steps of Theoretically Derived Distribution (TDD) of detention basin outflows.*

*(1) Hydraulic simulation with natural hydrograph: (1A) sketch of the detention basin section, composed of a low-level opening and a crest spillway, (1B) stage-storage capacity curve of the detention basin, (1C) t-Q plot of flood mitigation with natural hydrograph; (2) series of simulations with rectangular hydrographs: (2A) t-Q plot of flood mitigation with rectangular hydrograph, (2B) probability density function (pdf) of the outflows, theoretically derived from a Gumbel distribution of inflows , (2C) series of inflow maxima and outflow maxima for the numerical hydraulic simulation, (2D) Comparison between the Gumbel distributed inflows (solid black line) and the TDD outflows (dashed black line) and the numerical hydraulic simulation of inflows (red dots) and outflows (blue dots).*



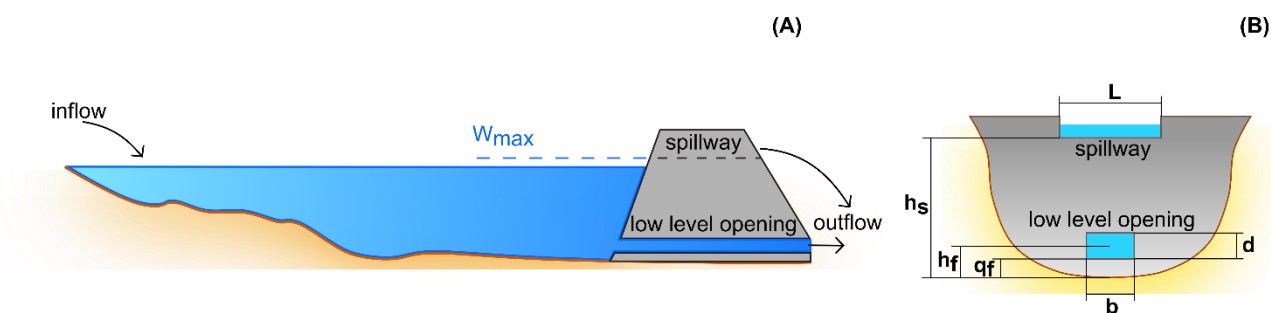

**Figure 1.** Sketch of a detention basin section with a basin capacity $W_{max}$, composed of a low-level opening, with an area equal to the product between $b$ and $d$, and a crest spillway of length $L$ and height $h_s$.









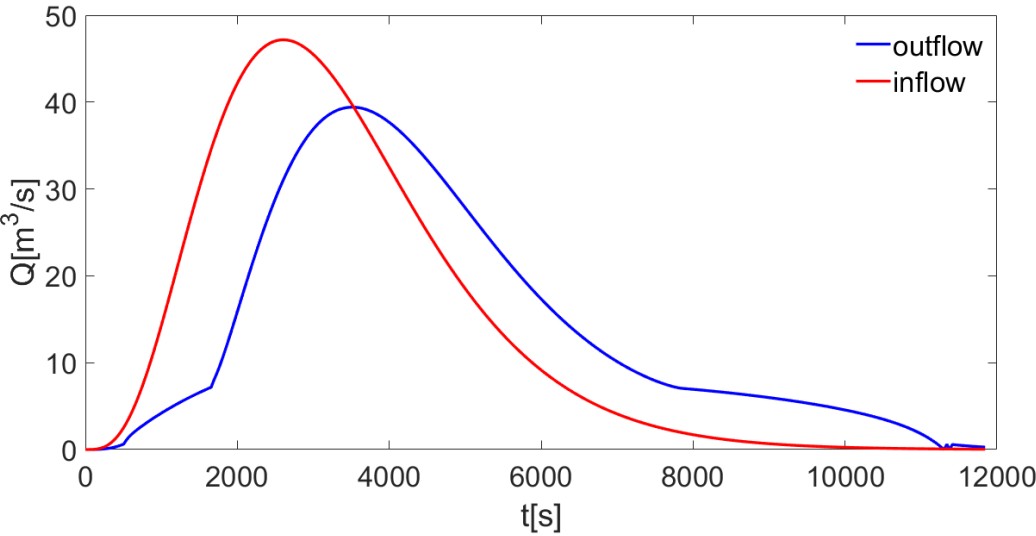

**Figure 2.** Example of flood mitigation induced by the presence of a detention basin obtained via numerical simulation. Other parameters are:   $b$=1m; $d$=1m; $n$=1.9; $h_f$=$q_f$+d/2; $h_s$ =4; $\mu_f$=0.85; $\mu_s$=0.385; $L$=3m, $W_{max}$=15,000.00 m$^3$


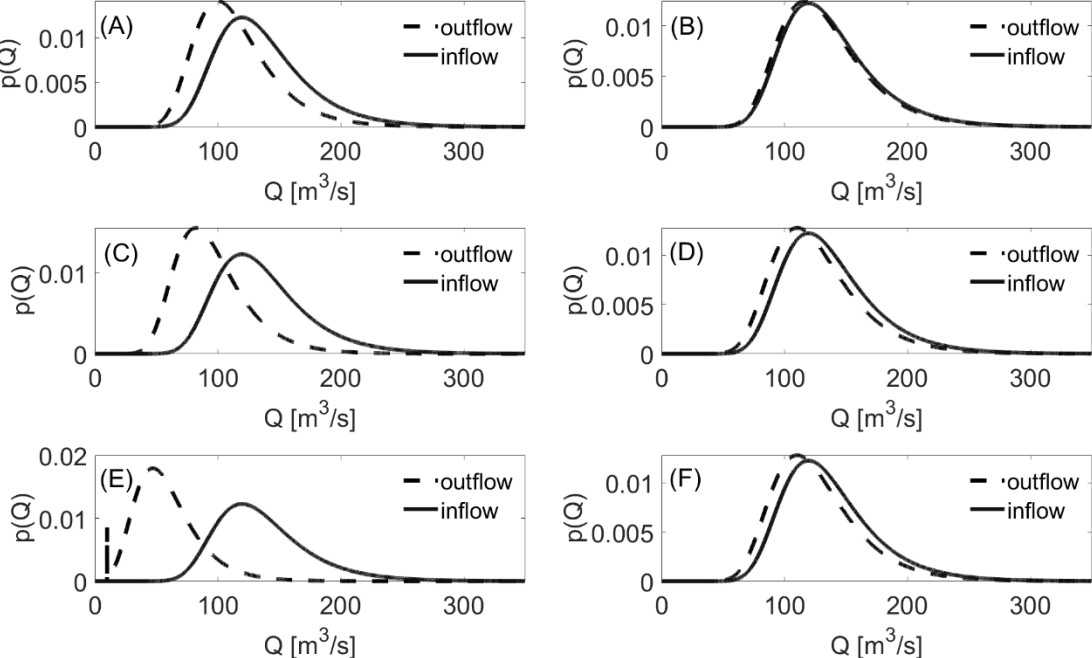

**Figure 3.** Derived probability density functions of the peak outflow obtained by modifying the height of the spillway crest ($h_s$) and the equivalent event duration $t_p$. Graphs on the left (**A-C-E**) are associated with a duration of half an hour and those on the right (**B-D-F**) to a duration of 1 hour, while elevation of the crest changes between 2 (**A, B**), 4 (**C, D**) and 8m (**E, F**). Other parameters are: α =30m³/s; β =120 m³/s; $b$=1m; $d$=1m; $n$=1.5; $h_f$=$q_f$+$d$/2=0.5m; $h_s$ =2; $\mu_f$=0.85; $\mu_s$=0.385; $L$=5m.



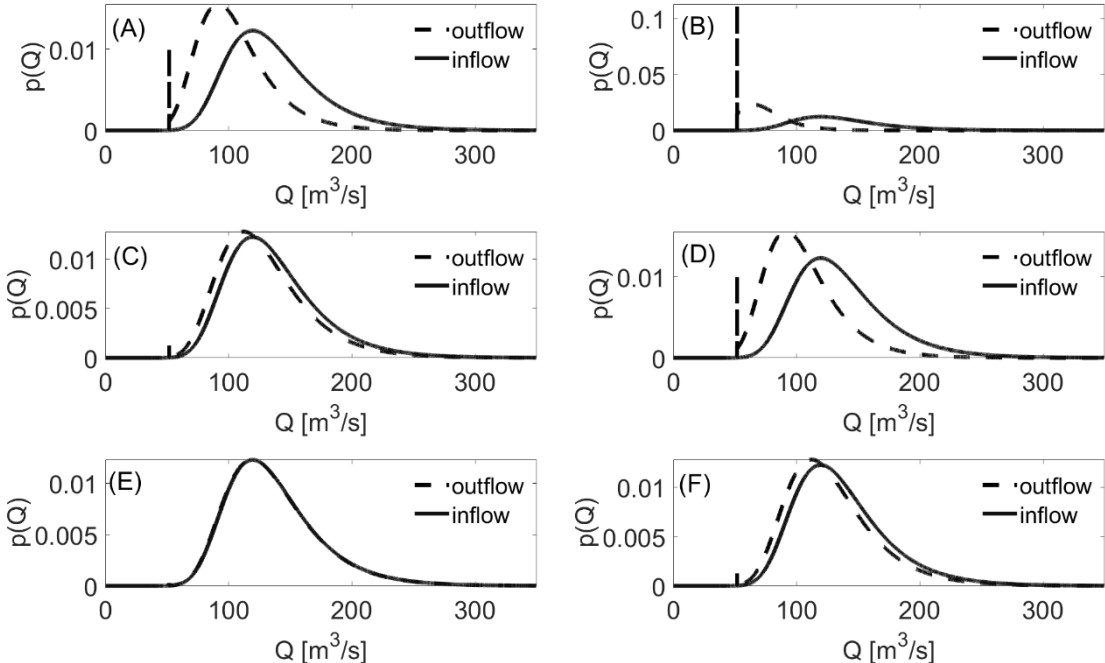

**Figure 4.** Derived probability density functions of the outflow associated with different equivalent event durations, $t_p$, using two different coefficients of the stage-storage capacity curve, $w_1$, which was set equal to 5000 in **A**, **C**, and **E**, while it assumes values of 10000 $m^3$ in **B**, **D**, and **F**. Event duration changes between 0.5 hours (**A, B**), 1 hour (**C, D**), and 2 hours (**E, F**). Other parameters are: $\alpha$ =30$m^3$/s; $\beta$ =120 $m^3$/s; $b$=4m; $d$=2m; $n$=1.5; $h_f$=$q_f$+$d$/2; $h_s$ =4; $\mu_f$=0.85; $\mu_s$=0.385; $L$=6m.



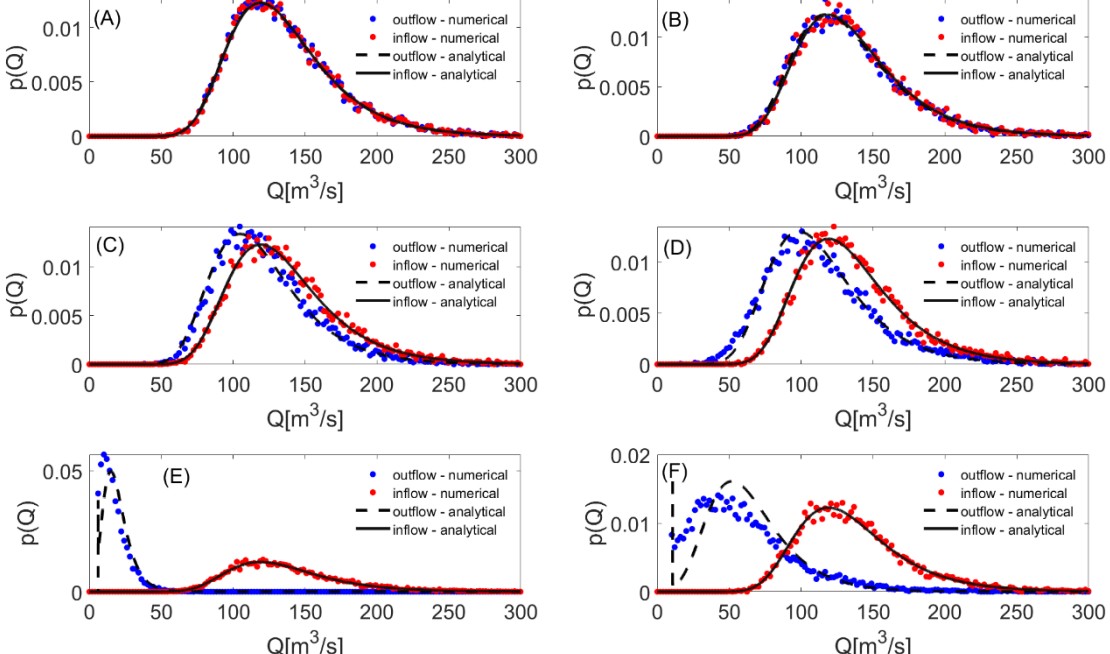

**Figure 5.** Comparison between derived probability density functions of the outflow (continuous black line for the inflow and dashed black line for the outflow) and the empirical pdfs obtained via numerical hydraulic simulation (red dots for inflow and blue dots for outflow). Graphs provide the following parametrizations: (**A**) $w_1$=2000; $h_s$=4m; (**B**) $w_1$=1500; $h_s$=9m; (**C**) $w_1$=5000; $h_s$=4m; (**D**) $w_1$=3000; $h_s$=10m; (**E**) $w_1$=40000; $h_s$=4m; (**F**) $w_1$=6000; =10m. Remaining parameters are: $\alpha$ =30m³/s; $\beta$ =120 m³/s; $b$=1m; $d$=1m; $n$=1.9; $h_f$=d/2; $\mu_f$=0.85; $\mu_s$=0.385 ; $L$=4m; $t_p$ = 2h.



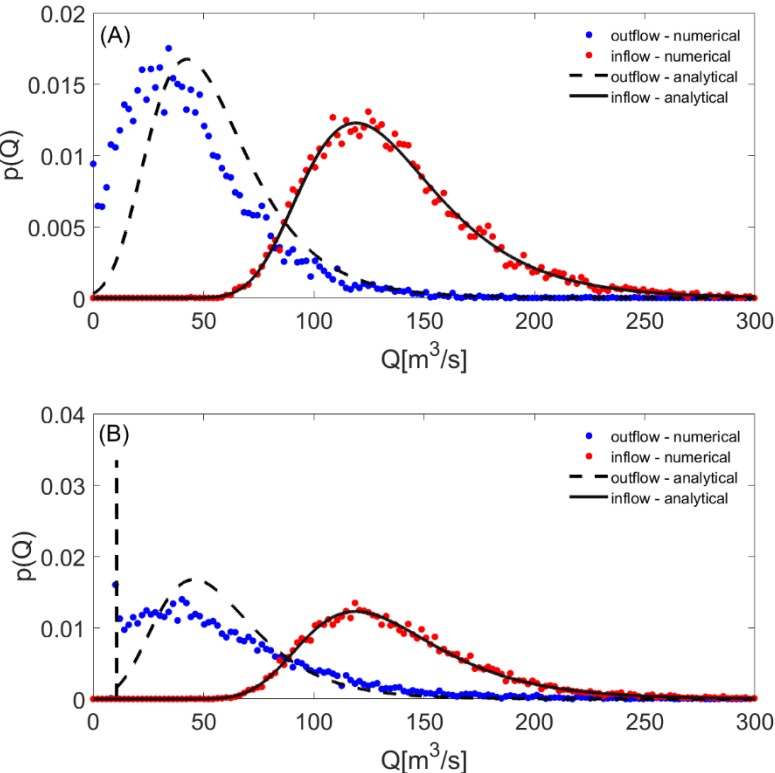

**Figure 6**. Comparison between derived probability density functions of the outflow (continuous black line for the inflow and dashed black line for the outflow) and the empirical pdfs obtained via numerical hydraulic simulation (red dots for inflow and blue dots for outflow) obtained by assuming the absence (**A**) or the presence (**B**) of the low-level opening. Remaining parameters are: $\alpha$ =30m³/s; $\beta$ =120 m³/s; $w_1$=6500; $h_s$=10m; $b$=1m; $d$=1m; $n$=1.9; $h_f$= $d$/2; $\mu_f$=0.85; $\mu_s$=0.385; $L$=4m; $t_p$ =2h.



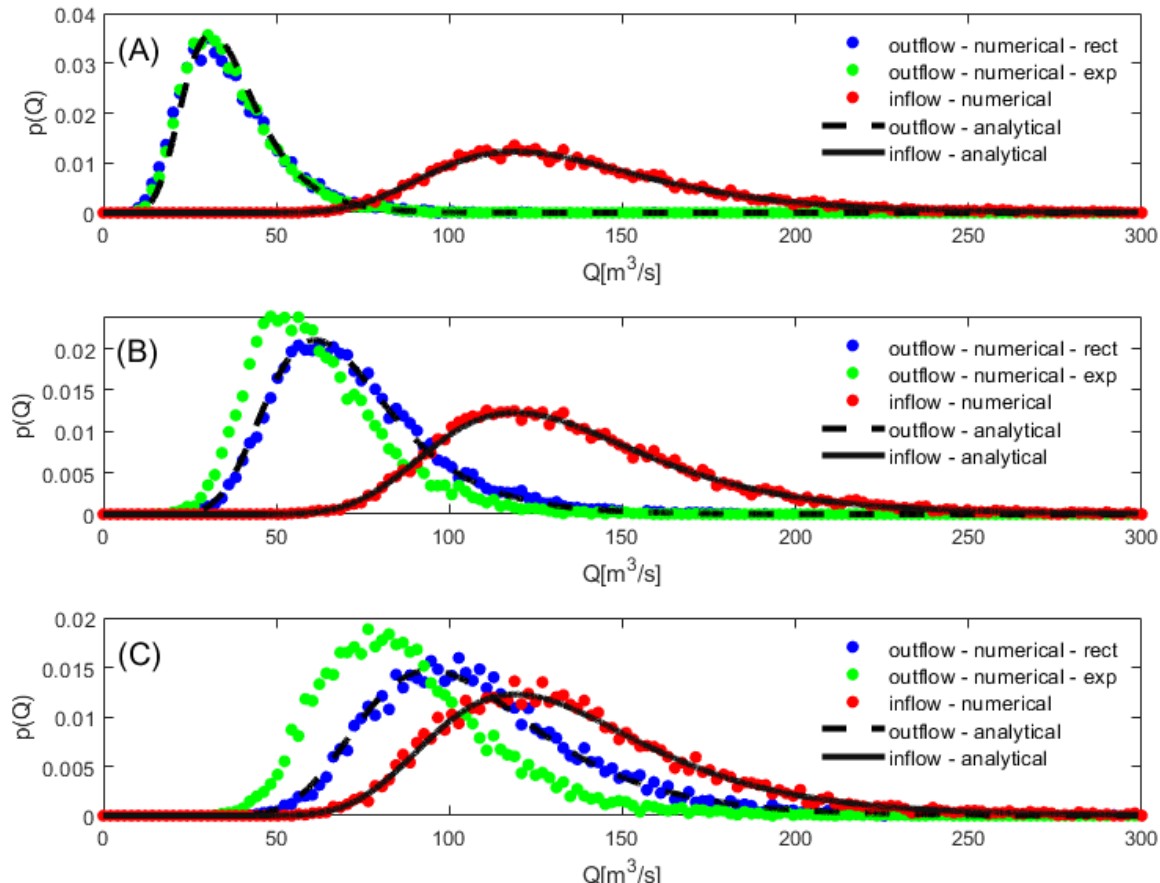

**Figure 7**. Comparison between derived probability density functions of the outflow (continuous black line for the inflow and dashed black line for the outflow) and the empirical pdfs obtained via numerical hydraulic simulation (red dots for inflow, blue dots for outflow obtained by incoming rectangular hydrographs and green dots for outflow obtained by incoming exponential hydrographs), assuming three equivalent event durations $t_p$ of half an hour (**A**), 1 hour (**B**) and two hours (**C**). Remaining parameters are: $\alpha$ =30m³/s; $\beta$ =120 m³/s; $w_1$=5000; $h_s$=4m; $b$=1m; $d$=1m; $n$=1.9; $h_f$= $d/2$; $\mu_f$=0.85; $\mu_s$=0.385; $L$=3m.
