# Peer review of "Impact of Detention Dams on the Probability Distribution of Floods"

_Hydrology and Earth System Sciences, 2021_

## Author Response (AR1)

**Response to Reviewer 1**

We would like to thank the first reviewer for the significative and constructive suggestions, that allowed us to improve the quality of the manuscript and clarify some concepts. We carefully considered all observations and reviewed the manuscript accordingly. Specific comments are addressed in the following using the same order adopted by the reviewer.

*RC: The authors are very honest stressing the strong simplifying assumptions used and this is clear throughout the text. Yet as reader I strongly felt that these assumptions and their effects should be discussed in more detail. Starting with equation 4 (I might be missing something here) it is not clear to me why Equation 3 cannot be used, meaning that is not much more complicated. Would the use of Eq.3 complicate that much the analytical calculations and make impossible the analytical formulations?*

AC: Basically, equations 4 and equations 3 differ only for the assumption that the outflow in eq. 4 is assumed equal to the inflow as long as h<hf. This assumption helps in the theoretical derivation of the probability distribution of the peak outflows and affect the shape of the distribution only for the low flows. Moreover, such an assumption may be considered reasonable for the scope of the present manuscript, which is more focused on the right tail of the derived probability distribution of the outflows.

*RC: Could you please offer more details on the impacts of the rectangular hydrograph assumption? Yes, it can significantly overestimate the flood volume but how much and under what conditions? How strong in the linearity assumption leading to the same exponent values (eq. 7)?*

AC: Rectangular hydrographs have been used for design purposes in several design applications. As demonstrated in the graph of figure 7, the assumptions is quite reasonable as long as the flood hydrograph has a relatively short duration. This means that the assumption can be used in small river basins with a lag-time of less than one hour.

*RC: The symmetric assumption leading to equation 10 how realistic can it be? It is well-known*
*if I am correct the volume is not symmetrically distribution around the peak.*

AC: The use of a synthetic hydrograph symmetric respect to the peak is an approximation which have a limited impact on the dynamic of the process. It is introduced to provide an estimate of the impact of non-uniform discharge on the lamination process. We could potentially use a different shape of the hydrograph, but this would have increased the number of modelling parameters, while this form represents the simplest form known.

*RC. Could you provide some extra details on the nature of the tails of the derived distribution in Eq 15? It is well accepted in the literature that floods peaks are described by heavy tails (see for example Vogel & Wilson (1996), Villarini and Smith (2010) and recently Miniussi et al. (2020) and Zaghloul et al. (2020)).*

AC: The choice of a Gumbel distribution is not mandatory in the proposed schematization. It is absolutely true that the Gumbel distribution is not necessarily the best option for the description of floods, but it represents a reference distribution for flood distributions. As we stated in the text, the choice of the flood distribution can be any of the available in the literature, because the derived laminated discharge can be used to obtain a derived distribution for any flood distribution chosen. We have included some examples of applications based on the assumption of floods distributed according to a Fréchet and a Weibull distribution.

*RC: Finally, I believe the readers would be very curious to see how the results would be modified if a heavy-tailed distribution was used instead of the Gumbel which has exponential tail. The exponential tails offer "no surprises" in the generation of random discharge values and thus the good results shown might be case specific and only for the Gumbel distribution. What would be the performance if really heavy tailed distributions were used, e.g., a GEV with shape parameter close to 0.5?*

AC: In order to satisfy readers' curiosity, we provide in the following few examples where the peak outflow distributions are derived using flood peaks distributed according to the three types of GEV distributions (Gumbel, Fréchet and Weibull). Reliability of the derived distribution has been tested using a numerical simulation, which demonstrates the good agreement with the theoretical functions.

Figure 1. Comparison between three different derived probability density functions of the peak outflows obtained using three different flood peak distributions and the empirical pdfs obtained via numerical hydraulic simulation (red dots for inflows, blue dots for outflows). The three graphs are obtained modifying the shape parameter, $\xi$, of the GEV distribution, which is equal to 0 for Gumbel distribution (A), 0.5 for Fréchet distribution (B) and - 0.5 for Weibull distribution (C). Remaining parameters are: the scale parameter of GEV $\alpha$ =30m$^3$/s; the location parameter of GEV $\beta$ =120 m$^3$/s; $w_1$=5000; $h_s$=4m; $b$=1m; $d$=1m; $n$=1.9; $h_f$= $d$/2; $\mu_f$=0.85; $\mu_s$=0.385; $L$=3m; $t_p = 1$h.

*RC. Please double check your equations, for example, Equation 13 is not correct, it should be dg^-1(y)/dy f(g^-1(y))*

AC: We thank the referee for this suggestion. The text has been modified accordingly.

[Figure]

**Response to Reviewer 2**

we would like to thank also the second referee for his/her effort and constructive suggestions, that allowed us to improve the quality of the manuscript. We have carefully considered all his/her observations and reviewed the manuscript accordingly.

*RC: I believe the reference to "flood control systems" in the title raises the expectations beyond what is presented in the paper, as there are other types of flood control systems that cannot be treated with the same mathematical framework proposed by the authors. I therefore recommend changing the title to refer more specially to "flood detention basins". The symbol D is not defined in the notation list. Given that the symbol q is used for discharge, I recommend using a different symbol for the height of the low-level opening instead of qf.*

**AC: We agree with the reviewer. Title will changed in … "Impact of Detention Dams on the Probability Distribution of Floods". We will also add the flood event duration *D* in the notation list. Finally, we will change *qf* in *lf*.**

*RC. P80,90: Use italic style for mathematical variables.*
*P10: "the undisturbed flood distribution is assumed to be Gumbel distributed" => "the undisturbed flood peaks are assumed to be Gumbel distributed"*
*P50: "see i.e., Manfreda et al." => "e.g." not "i.e."*
*P165: "... mathematically inverting Equation..." => "mathematically by inverting Equation..."*
*P140: "computed comparing... and setting..." => "computed by comparing... and imposing..."*
*P255: Remove "realized"*
*P235: "the impact due to the approximation adopted by the rectangular hydrographs" =>*
*I suggest changing this as follows: "the impact of the assumption of rectangular inflow hydrograph"*
*P235: "allowed to reproduce correctly the flood mitigation that looks very similar to those..." => "produced probability distributions of the outflow that look very similar to those...".*
*L375: 15,000.00 => 15,000*

**AC: All these suggestions have been already implemented in the revised version of the manuscript.**